# Physical and Motor Fitness Tests for Older Adults Living in Nursing Homes: A Systematic Review

**DOI:** 10.3390/ijerph19095058

**Published:** 2022-04-21

**Authors:** Luis Galhardas, Armando Raimundo, Jesús Del Pozo-Cruz, José Marmeleira

**Affiliations:** 1Departamento de Desporto e Saúde, Escola de Saúde e Desenvolvimento Humano, Universidade de Évora, Largo dos Colegiais, 7000-727 Évora, Portugal; ammr@uevora.pt (A.R.); jmarmel@uevora.pt (J.M.); 2Comprehensive Health Research Centre (CHRC), Palácio do Vimioso, Gabinete 256, Largo Marquês de Marialva, Apart. 94, 7002-554 Évora, Portugal; 3Department of Physical Education and Sports, University of Seville, 41013 Sevilla, Spain; jpozo2@us.es; 4Epidemiology of Physical Activity and Fitness across Lifespan Research Group (EPAFit), University of Seville, 41013 Sevilla, Spain

**Keywords:** assessment, long-term care facilities, measure, older adults, performance-based testing

## Abstract

This systematic review aimed to identify the physical/motor fitness tests for nursing home residents and to examine their psychometric properties. Electronic databases were searched for articles published between January 2005 and October 2021 using MeSh terms and relevant keywords. Of the total of 4196 studies identified, 3914 were excluded based on title, abstracts, or because they were duplicates. The remaining 282 studies were full-text analyzed, and 41 were excluded, resulting in 241 studies included in the review. The most common physical component assessed was muscle strength; 174 (72.2%) studies assessed this component. Balance (138 studies, 57.3%) and agility (102 studies, 42.3%) were the second and third components, respectively, most widely assessed. In this review, we also describe the most used assessment tests for each physical/motor component. Some potentially relevant components such as manual dexterity and proprioception have been little considered. There are few studies assessing the psychometric properties of the tests for nursing home residents, although the data show that, in general, they are reliable. This review provides valuable information to researchers and health-care professionals regarding the physical/motor tests used in nursing home residences, helping them select the screening tools that could most closely fit their study objectives.

## 1. Introduction

Currently, most people can expect to live to over 70 years old [1]. Thus, the increase in average life expectancy combined with the sharp decline in fertility rates is leading to the rapid ageing of populations worldwide [2]. An important question related to the increase in longevity is the relationship between older ages and the person’s general health. With advancing age, there is a progressive deterioration in the physical and mental health of the elderly and a consequent increase in the need for greater medical and social assistance [3].

When talking about health throughout the natural ageing process, it is assumed to be multifactorial, resulting from a continuous interaction between genetic and environmental influences, which makes the group of older people quite heterogeneous [3,4]. The ageing process leads to inevitable life changes and is characterized by a progressive decline in physiological functions [5]. Older adults often show low aerobic capacity, low muscle strength, balance limitations, and other physical/motor limitations [6,7]. Regular physical activity and/or physical exercise are essential for healthy ageing and contribute to better mental health [6]. It can help prevent, delay, or manage many costly and challenging chronic diseases handled by older adults, especially those living in nursing homes. It can also decrease the risk of moderate or severe functional limitations in older adults and the risk of premature death [8]. Assessment methods are needed to assess these capabilities.

In this review, we focused on older adults living in nursing homes, a group of the population that was characterized by a high proportion of very much older adults, most of them with marked levels of frailty [9,10]. The assessment of the physical and motor capacities of nursing home residents is of importance for health personnel to design proper (and individualized) intervention programs, monitor a person’s progress, and evaluate the effects of the interventions carried out [11]. To the best of our knowledge, in recent years, no literature review has focused on physical/motor assessment methods for people living in long-term facilities. From our perspective, knowledge of the most used assessment tests and the physical parameters most screened could be very pertinent to health professionals and researchers.

The principal aim of this systematic literature review is to highlight and categorize the most common tests used in recent years for assessing the physical function and motor skills of older adults living in institutionalized contexts. This study also intends to examine the psychometric properties of the tests. This information could help researchers and technicians to select the methods that best suit their objectives, whether to analyze the effects of treatments or to characterize the person’s abilities.

## 2. Materials and Methods

A systematic literature review of studies involving older adults, residents of nursing homes, or similar, and to which physical or motor assessment methods were applied, was performed. The Preferred Reporting Items for Systematic Reviews and Meta-Analyses (PRISMA) 2020 statement [12] was applied to improve the report of this systematic review.

We registered the review in the PROSPERO International prospective register of systematic reviews (registration number CRD42020212338), and it is available at the following link: https://www.crd.york.ac.uk/prospero/display_record.php?ID=CRD42020212338 (accessed on 1 November 2020).

### 2.1. Data Sources and Search Strategy

Electronic databases (PubMed, Scopus, and ScienceDirect) were initially searched for articles published between January 2005 and 31 December 2020. Subsequently, a new search was conducted for articles published until 31 October 2021. Using MeSh terms and relevant keywords the following semantic categories were entered: ‘nursing homes’, ‘institutionalized’, ‘residential care facility’, ‘long-term facility’, ‘aged, 65 and over’, ‘older’, ‘elderly’, ‘older adults’, ‘physical’, ‘motor’ ‘physical’, ‘tests’, ‘physical assessment’, ‘motor tests’, and ‘motor assessment’. We also screened the reference lists of review articles. The review was conducted using the DistillerSR (Evidence Partners Incorporated, Ottawa, Canada), an online specialized systematic review software. All identified citations were uploaded to DistillerSR, and duplicates were identified and removed.

### 2.2. Inclusion and Exclusion Criteria

Studies were eligible if they met the following inclusion criteria: (1) the participants were institutionalized (nursing homes or similar); (2) the participants were on average older than 65 years; (3) physical or motor tests (assessment methods) were used to measure specific abilities/skills; and (4) written in English. Studies were excluded if (1) the publication was a systematic review, abstract, study protocol, letter, commentary, a study reporting only qualitative data, dissertation, or poster abstract; (2) the work was published before 2005; or (3) they just applied scales or questionnaires, with indirect evaluation or by interview, very common, for example, in activities of daily living (ADL) assessment. It should be noted that we only consider methods/assessments that directly assess performance in pre-established activities.

### 2.3. Selecion Process

After the literature search, an initial selection of studies was performed according to their titles, followed by a selection after reading the abstracts. One reviewer (LG) performed both steps to identify those studies that met the inclusion criteria. After that, another reviewer (JM) checked whether all was in accordance. Disagreement was solved with full-text screening by all the research team.

In the next step, a full-text analysis was performed to check whether the studies identified previously matched with the inclusion criteria. Subsequently, all the research team hand-searched for other studies that were not already found in the initial literature search. Finally, full-text analysis and data extraction from the final selected studies were performed.

### 2.4. Data Extraction

The data extracted from the selected studies included: year of publication; type of study; mean age of population sampled; sample size; assessment methods (physical or motor); and abilities/skills that were measured. We also collected, when studies mentioned them, which psychometric characteristics were analyzed and the respective statistics.

### 2.5. Study Quality

We evaluated the quality of the studies that examined the psychometric properties. To the best of our knowledge, only one tool—the COnsensus-based Standards for the selection of health Measurement INstruments (COSMIN) Risk of Bias checklist [13,14,15] (www.cosmin.nl/, accessed on 1 November 2021)—is proper for assessing the methodological quality of outcome measurement instrument [16]. From this tool, we selected and applied, adjusting to the individuality of each study, the items’ content validity, structural validity, internal consistency, cross-cultural validity/measurement invariance, reliability, measurement error, criterion validity, hypotheses testing for construct validity, and responsiveness.

For the other studies, as our interest was only to understand what tests the scientific community uses most to assess physical or motor abilities/skills in nursing home residents, we did not evaluate their quality. For these studies, we did not collect any outcomes (tests scores), only the tests that were used. This is in line with previous studies, e.g., [17,18], that focused on the health benefits of sports and physical activity for older adults and on mental health. To comprehensively review the physical/motor fitness tests for nursing home residents, we included studies with various designs (e.g., observational, randomized controlled trial and controlled non-randomized). One should note that this search was carried out in the most reputable databases, assuming the quality of the studies.

## 3. Results

### 3.1. Selection Process

Of the total of 4196 studies, 3794 were excluded based on title or because they were duplicates. Of the remaining 402, 120 were excluded based on title and abstracts. For the remaining 282 articles, the full text was analyzed. In this phase, 41 additional articles were excluded, resulting in a total of 241 articles (involving 27,646 older people) to include in the review. The process and outcome of the literature selection are presented in detail in Figure 1.

### 3.2. Categorization of the Physical and Motor Assessment Methods

To frame each identified assessment method into categories, we use the following definitions.
Muscular Strength—the ability to exert a force on an external object/target or resistance [19].Balance—maintaining the position of the body’s center of gravity vertically over the base of support. Relies on rapid, continuous feedback from visual, vestibular, and somatosensory structures [20].Agility—rapid whole-body movement with a change of velocity and/or direction [21].Gait—walking patterns as a combination of different joints and muscles active simultaneously to maintain an upright posture and to produce forward propulsion of the whole-body [22].Aerobic capacity—ability to use large muscles (muscle groups) in dynamic and moderate-to-high-intensity exercise for prolonged periods. Depends on the functional state of respiratory, cardiovascular and muscular systems [23].Flexibility—static flexibility is the degree to which a joint can be passively moved to the end-points in the range of motion; dynamic flexibility is the degree to which a joint can be moved as a consequence of a muscle contraction [24].Reaction time—simple reaction time is the time needed to react to a single stimulus, whereas choice reaction time refers to the time needed to make a choice and respond to one of a number of possible stimuli [25].Activities of Daily Living (ADL)—refers to a large scope of daily activities required to complete a day. ADL can include self-care tasks and more complex instrumental tasks such as shopping and food preparation [26].Dexterity—voluntary movements, with the hands, used to manipulate small objects during a specific task [27].Proprioception—the sense of the position of the body and body parts in the space, including body segment static position, displacement, acceleration, velocity, and muscle sense of tension/effort [11].Paratonia—an external stimulus-dependent increase in muscle tone that is absent at rest [28].

The most common physical/motor component assessed was muscular strength (174 studies); of the 241 articles included in this review, 72.2% included methods that assess this component. Balance, agility, and gait were also frequently assessed in the target population.

In Table 1, it is possible to verify which physical/motor components we identified in the review studies, evaluated in recent years, as well as their frequency of use. 

Table 2 describes the test batteries identified in the reviewed studies and their frequency of use. We found that the Short Physical Performance Battery, the Berg balance scale, and the Tinetti test are the three most popular test batteries. 

Table 3 shows all the tests found in this literature review, organized by components. To analyze the tests individually, the batteries that assessed different physical and/or motor components (e.g., balance and strength) were separated ‘test-by-test’. The batteries dedicated to one component (e.g., Berg balance scale) were considered as an individual test. Based on this criterion, we identified 97 different tests, involving the various physical and motor characteristics of older nursing home residents. For each category, the tests are presented in descending order of the number of studies in which they were used.

Table 3 shows that there are tests that stand out for their high frequency of use over the past 15 years. 

To assess strength, we identified five stand-out tests, widely referred to in the reviewed studies: handgrip strength test (handgrip dynamometer), which measures the maximum isometric strength of the hand and forearm muscle; five times sit-to-stand test, a simple and rapid method for quantification of lower extremity muscle strength [266]; 30 s sit to stand test [267], a simple test to assess the lower-limb muscle strength; handheld dynamometer tests, which are electronic devices (several brands) used to assess strength in different muscles/movements; and the arm curl test, also a simple and rapid method that assesses the upper body strength [267]. We also identified another 16 assessment methods used less frequently.

As described previously, balance is the capacity to maintain the position of the body’s center of gravity vertically over the base of support [20]. We identified 20 different tests to assess this physical capacity. The most used tests were the tandem test, semi-tandem test, and feet together test [268]. It is equally important to highlight that the Berg balance scale [269] and the Tinetti test [270] are also frequently used. Both test batteries include several tasks to generate a person’s composite balance score.

In the agility component, the most frequent method used was the timed up and go (TUG) test, which measures how long it takes for a person to stand up from a chair, walk three meters, go around a cone, walk back to the chair and sit again [271]. This assessment method was applied in 82 different studies involving 7212 nursing home residents. Although we included the TUG in the agility component as it is a method in which there are different gait speeds and there is a change of direction, this method is also associated with a vast set of other physical/motor components such as balance, gait speed, and in general, functional mobility [272,273].

We identified 19 tests to access gait. The tests are relatively similar, with the major difference being the walking distance. It is important to highlight that the most used method is the gait speed test included in the short physical performance battery, which measures the lower extremity physical performance [274], and is widely used, as can be seen in Table 2. In the gait speed test, the person’s usual gait speed is evaluated (including with a walking aid), and the person must walk the stipulated distance at their usual walking speed [274]. The score of this test is the time the person needs to walk the distance.

The aerobic capacity is another important physical component, for which we identified five different tests. The most common method is the six-minute walk test [275], applied in 30 studies (involving 2486 persons). This assessment method, based on the distance covered over a time of six minutes, is a sub-maximal exercise assessment used to measure endurance and aerobic capacity [275]. 

To assess flexibility, we identified seven different methods. Four methods are clearly preferred: the back-scratch test [267], the chair sit and reach test [267,276], the goniometric measures, and the sit and reach test [277]. All these tests are quick and easy to apply and do not require complex materials to be purchased.

We also found some physical/motor components that included a few tests, namely reaction time, activities of daily living, dexterity, proprioception, and paratonia.

Throughout our research, we identified some studies that analyzed the psychometric properties of the assessment methods. To analyze the quality of these studies, we used the COSMIN Risk of Bias checklist (www.cosmin.nl/, accessed on 1 November 2021). 

Most of the studies that assessed the psychometric properties presented adequate and/or better methodological quality according to the COSMIN Risk of Bias checklist (Table 4).

In the studies presented in Table 5 that evaluated the psychometric properties, the assessment tests or the batteries applied, presented, in general, at least acceptable reliability, validity and measurement accuracy.

## 4. Discussion

Our primary aim with this review was to provide valuable information to the researchers or healthcare professionals regarding the physical/motor tests that are common in nursing home residences, helping them select the screening tools that could most closely fit their study objectives. Overall, we found that there is a wide set of physical/motor tests that have been widely used over the last few years for assessing several components among older people living in nursing homes. However, there are assessment methods and physical and motor components that are clearly preferred over others.

In this review, we identified 97 different methods that covered the domains of strength, balance, agility, gait, aerobic capacity, flexibility, reaction time, dexterity, proprioception, activities of daily living, and paratonia. We found that six physical components are the most prominent in scientific research, namely strength, balance, agility, gait, aerobic capacity, and flexibility.

In each component, there is a large heterogeneity in the tests that have been used, although some of them seem to be more relevant. For the assessment of strength, the physical component most analyzed, we identified 21 different tests. Five of these tests were used more frequently, as each one was applied in at least 10% of the 241 studies included in this review. The most common assessment method is the hand handgrip strength test, which was used in 97 of the 241 studies (~40%). This test measures the amount of static force that the hand can squeeze around a dynamometer [278] and has a high relationship with the overall strength (relative strength to body mass) [29]. Handgrip strength also influences dependency on daily functioning and the quality of life [138]. To assess the strength of the lower limbs, the most used assessment method is the five times sit to stand test [274], used in 47 studies (~20%). This assessment method is rapid and easy to apply (with accessible materials), it is associated with the person’s ability to walk and is also related to the risk of falls in nursing home residents [73]. In the five times sit to stand test, the time it takes the person to get up and sit down in a chair five times is evaluated, the score is the time needed to perform the task [274]. Additionally, the 30 s sit to stand test [279], the handheld dynamometer tests (evaluates the strength of various muscles), and the arm curl test are methods used more recurrently, being present in at least 10% of studies. In a study carried out in a nursing home, the 30 s sit to stand test had an excellent relative inter-rater reliability, supporting its potential for clinical use [63]. For the other methods, we did not identify data related to their reliability. The remaining 16 assessment tests we identified have low application rates, below 10% in the 241 studies selected for this review. In general, we found tests that assess muscle strength, but not muscle mass. Some data show that a significant and high association exists between muscle strength and physical performance in weak older adults, and that the clinical approach for weak or frail older adults should focus on muscle strength rather than muscle mass [280]. It should be noted that there is a high prevalence of frailty among nursing homes residents [9]. Still, in community-dwelling older people (aged 80 years or older), some data point that, for instance, the calf circumference may be positively related to a lower frailty index and higher functional performance [281].

Balance is the second component that researchers have given the most relevance, and 57% of the reviewed studies include balance tests (Table 1). We found 20 assessment methods included in this component. The most used tests are the tandem test, the semi-tandem test, and the feet together test [268]; one or more of these tests was applied in over 16% of the studies. These methods are relevant when the aim is to analyze the static balance of elderly people and/or their postural stability [182,198]. They are quick and easy to apply, do not require expensive materials, and assess the capacity to maintain the position of the body’s center of gravity vertically over the base of support [20]. In the balance component, it is also important to highlight two assessment batteries, the Berg balance scale [269] and the Tinetti test [270], as both are part of the top three batteries, used in approximately 15% of the reviewed studies. As mentioned previously, the Berg balance scale [269,282] assesses balance, involving 14 different tasks scored between 0 and 4, in which higher scores represent better performance. We found that the Berg balance scale shows excellent relative inter-rater reliability, good test–retest reliability, high internal consistency, and enables the identification of fall status in nursing home residents [63,207]. Additionally, a literature review showed that this scale presents high intra-rater and inter-rater reliability, and high absolute reliability [282]. The Tinetti test (score from 0 to 40) is a simple clinical balance assessment that measures conditions associated with falls. This test is divided into two parts. One assesses static balance with 14 items (scored from 0 to 24), and the other assesses dynamic balance with 10 items (scored from 0 to 16) for a total score out of 40; a higher score reflects a better performance [270]. The Tinetti test also shows good inter-rater reliability in nursing home residents [222]. The data highlight the potential of these two test batteries for research and clinical use.

We assume agility to be a rapid whole-body movement with a change of velocity and/or direction in response to a stimulus [21]. Based on this definition, we include in this component two widely used assessment methods, namely the TUG [271] and the eight-foot up and go test [267]. Of these methods, the TUG has been the most used in recent years, appearing in 82 articles (34% of the reviewed studies). The explanation for the high frequency of use of this method could be, as previously described, that it is easy to apply, with no overly specific materials, and it is a method that presents excellent reliability and acceptable measurement precision in nursing home residents [11]. Moreover, it is associated with the risk and history of falls [283] and is also widely used for elderly people living in the community [284,285].

To analyze/assess gait, we found 19 methods, although some of them use very similar protocols. It is important to highlight that 15 methods were used only in ≤7 articles (3% of those reviewed). The remaining four methods were used more regularly. The most applied assessment method for gait is the gait speed test-SPPB, used in 37 studies. This test is part of the Short Physical Performance Battery, which was the test battery most frequently used (Table 2), and assesses physical performance in the components of balance, gait, and strength [274]. This test seems to have high relative and absolute reliability in nursing home residents [286], which strengthens its potential for clinical and research use. The remaining three tests most used in recent years for assessing gait have protocols similar to each other, but using different walking distances (4, 6, and 10 m walking tests) [287,288,289]. All these methods access the person’s usual gait speed. The person is instructed to walk a predetermined distance at their usual speed. The usual gait speed is estimated by dividing the distance by the time. [198]. To the best of our knowledge, in the last 15 years, only the psychometric properties of the six-meter walking test method have been studied, and the results reveal an excellent relative inter-rater reliability and an absolute reliability of 0.08 [63]. The gait speed is an indicator of ADL function [290], and gait ability can predict functional decline, falls, and disability in nursing home residents [291]. 

Aerobic capacity depends on the functional state of the respiratory, cardiovascular, and muscular systems [23]. These physical components were studied in 17% of the reviewed studies. We identified five different assessment tests, but just one method was most applied in the past 15 years—the six-minute walk test [275]—which was applied in 30 studies (12.5%). It is simple to apply, requiring only ample space, a measuring tape, cones and a stopwatch. The score of this test is the distance (in meters) covered over six minutes [275]. We did not find studies on the psychometric properties of this assessment method focused on people living in nursing homes. Nevertheless, there are some psychometric results regarding older adults that are frail or have specific disabilities. Thus, a study carried out with frail older adults in daycare and residential care facilities (mean age 87.1), reported that the six-minute walk test showed excellent test–retest and inter-rater reliability, correlating moderately with other functional measures [292]. Studies with community-dwelling older adults with Parkinsonism [293], and with Alzheimer’s disease [294], also reported excellent test–retest reliability of the six-minute walk test. 

Another physical component that has also received attention from researchers in recent years is flexibility. We have identified seven different methods to assess this ability, two of which were used more regularly, namely the back-scratch test [267], applied in 18 of the reviewed studies (7.5%), and the chair sit and reach test [267,276], applied in 16 of the reviewed studies (6.6%). As in other methods mentioned above, the protocols are simple, of rapid application, and use accessible materials. It should be noted that flexibility considered in nursing home residents [294] is relevant, as it is an important functional skill for the participants to carry out daily activities, such as combing their hair, changing clothes or washing their bodies during a shower [295]. Considering the importance of this ability in some activities of daily living, it should be more considered in the future both at a clinical and research level.

The remaining physical components that we identified in this review were least evaluated, namely ADL, reaction time, manual dexterity, proprioception, and paratonia. Regarding ADL, it is important to highlight that we only collected information from methods that directly assess this component, that is, which assesses the person performing the tasks. ADLs are frequently evaluated through questionnaires or scales [200,296,297], without observing the person performing any type of activity. According to some data, older adults tend to underestimate their capabilities in self-reported measures [298], so the direct observation of ADL and instrumental ADL performance could overcome the potential bias associated with questionnaires. Eventually, it will be necessary to give more attention to this aspect and to look for evaluation methods that assess the actual performance of people living in nursing homes.

Reaction time is also another undervalued component, as we only identified three assessment methods, included in five studies. The reaction time assessment may be important in nursing home residents. Previous studies conducted in nursing homes and senior residences reported that reaction time could predict the risk of falling and the ability to solve complex cognitive tasks by indexing IADLs [299]. We suggest that in future studies, more attention should be given to this ability.

The hand and upper-limb function, including manual dexterity and proprioception, are crucial to maintaining independence and competence in performing ADLs. Nevertheless, there is an age-related decline of the upper limb and hand function [300], eventually supported since the nursing home residents receive 24/7 assistance, which reduces the activities independently performed. Manual dexterity can be an objective measure of a person’s skill to accomplish the ADLs required for independence [301]. For example, dexterity tests can help identify patients unable to perform adequate oral self-care [302]; it may also be a component of interest to be considered in future studies. Some studies have highlighted the importance of proprioception in older people, revealing, for instance, its importance to body stability and the prevention of falls [303,304]. However, the present systematic review shows that just three studies assessed this ability in nursing home residents, using four different assessment methods, two related to the upper limbs and the other two to the lower limbs. Regarding the upper limb, both tests—the arm ruler positioning test and the weight detection test—seem to have good relative reliability and acceptable measurement precision, showing potential for clinical and research use [11]. In our point of view, more attention should be given to manual dexterity and proprioception in nursing homes, as they can affect a person’s quality of life and autonomy.

Paratonia was the last component identified in this review and was assessed in one study by the paratonia assessment instrument (Table 3). This study analyzed the reliability and validity of this method, evidencing that it is a reliable and valid test and can be applied easily in daily practice [265]. Paratonia is a disorder generally associated with dementia, and it may be an instrument to be considered in nursing home residents as a way of screening for this medical condition [265].

## 5. Limitations

Some limitations of this review should be considered. Only studies published in English were included in this review, which could have limited the availability of some studies, and changed the frequency of use of some assessment methods. Another limitation is that the quality of some articles was not considered. However, our aim was only to understand which tests are most used by the scientific community to assess physical or motor abilities/skills in nursing home residents. Additionally, one should note that this search was carried out using the most reputable databases.

The present study also has some strengths. To the best of our knowledge, it is the first review focused on this topic, providing relevant evidence for older adults in nursing homes on (i) the most used physical and motor tests; (ii) the most assessed physical and motor components; and (iii) the psychometric-related data. Taken together, the data presented here can be very useful for clinicians and researchers.

## 6. Conclusions

This review included 241 studies, involved 27,646 older nursing home residents, and mapped the large heterogeneity in motor and physical functioning assessment methods applied. We identified 97 different tests, involving the different physical and motor characteristics. According to the data collected, we found that the most common physical/motor component assessed was muscular strength. We identified five stand-out tests to assess strength: the handgrip strength, the five times sit-to-stand, the 30 s sit to stand and the arm curl test. Additionally, balance, agility and gait were commonly assessed in the target population. For these components, the most used methods were the tandem test, semi-tandem test, and feet together test (balance); the timed up and go test (agility); and the gait speed test-SPPB (gait).

We suggest that more attention should be paid to other physical and motor aspects (e.g., reaction time and proprioception), as they also contribute to a better quality of life, better autonomy, and a healthy and active ageing. Furthermore, since information on the tests’ psychometric quality seems to be, in several cases, insufficient or absent, more related studies are needed in the near future.

This review provides valuable information to researchers and/or health-care professionals regarding the physical/motor tests that are common in nursing home residences, helping them select the screening tools that could most closely fit their study objectives.

## Figures and Tables

**Figure 1 ijerph-19-05058-f001:**
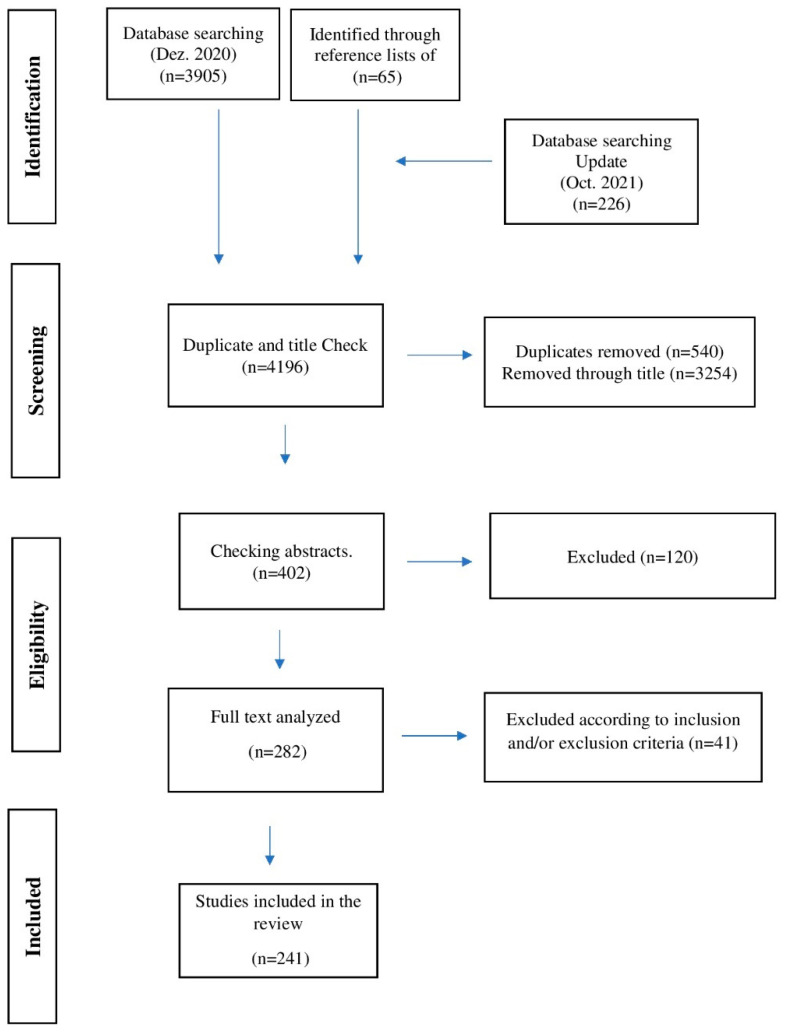
Flow diagram of database search.

**Table 1 ijerph-19-05058-t001:** Physical/motor components that were identified in the reviewed studies.

Components	Number of Studies	% of the Total	References
Strength	174	72.2%	[6,10,29,30,31,32,33,34,35,36,37,38,39,40,41,42,43,44,45,46,47,48,49,50,51,52,53,54,55,56,57,58,59,60,61,62,63,64,65,66,67,68,69,70,71,72,73,74,75,76,77,78,79,80,81,82,83,84,85,86,87,88,89,90,91,92,93,94,95,96,97,98,99,100,101,102,103,104,105,106,107,108,109,110,111,112,113,114,115,116,117,118,119,120,121,122,123,124,125,126,127,128,129,130,131,132,133,134,135,136,137,138,139,140,141,142,143,144,145,146,147,148,149,150,151,152,153,154,155,156,157,158,159,160,161,162,163,164,165,166,167,168,169,170,171,172,173,174,175,176,177,178,179,180,181,182,183,184,185,186,187,188,189,190,191,192,193,194,195,196,197,198,199,200]
Balance	138	57.3%	[10,11,29,32,33,36,38,40,43,46,47,49,50,51,52,58,59,60,62,63,64,66,70,71,74,77,78,79,82,84,85,86,87,90,91,92,93,94,96,97,98,100,101,103,104,105,107,108,109,110,111,117,126,130,131,135,136,138,140,141,142,145,146,147,148,149,150,151,152,153,154,155,156,157,167,168,171,175,179,180,181,182,183,184,185,188,189,192,194,198,201,202,203,204,205,206,207,208,209,210,211,212,213,214,215,216,217,218,219,220,221,222,223,224,225,226,227,228,229,230,231,232,233,234,235,236,237,238,239,240,241,242,243,244,245,246,247,248]
Agility	102	42.3%	[6,11,29,31,32,35,45,46,49,52,55,57,62,64,65,66,70,73,74,76,78,79,81,86,87,90,92,93,96,97,102,105,107,108,109,115,117,121,136,140,145,146,147,150,151,153,155,158,159,161,162,163,165,167,171,178,182,183,185,191,192,194,196,198,202,203,204,205,208,209,210,211,214,215,216,219,220,221,223,224,225,226,227,228,229,230,233,235,237,241,243,246,248,249,250,251,252,253,254,255,256,257]
Gait	96	39.8%	[10,29,30,32,33,35,36,37,38,39,40,43,47,48,49,50,51,53,54,57,58,59,60,62,63,64,65,68,71,73,74,77,78,79,80,82,84,86,94,96,98,103,104,105,106,107,110,111,112,126,129,130,131,132,133,135,136,141,142,144,145,147,149,150,151,153,155,156,158,163,166,171,175,177,180,181,182,184,187,192,195,198,216,217,219,227,228,230,239,241,246,248,249,258,259,260]
Aerobic capacity	41	17.0%	[6,31,43,46,55,79,80,81,87,93,95,116,117,119,120,125,136,146,147,148,150,151,161,162,163,164,165,167,173,178,181,182,186,189,191,193,202,243,259,261,262]
Flexibility	33	13.7%	[43,55,61,65,73,79,81,82,90,100,101,108,109,111,138,147,159,160,161,162,165,168,170,178,179,188,191,194,213,226,246,263,264]
Reaction Time	5	2.1%	[6,43,100,109,147]
Dexterity	4	1.7%	[100,138,157,179]
Proprioception	3	1.3%	[11,108,194]
ADL	1	0.4%	[84]
Paratonia	1	0.4%	[265]

Notes: ADL: activities of daily living.

**Table 2 ijerph-19-05058-t002:** Batteries of physical and motor tests applied in the reviewed studies.

Test Battery	Assesses	N (Studies)	Sample (Females/Males)	Mean Age	References
Short physical performance battery	Gait speed, chair stand and balance	37	6469 (4541/1928)	83.7	[10,29,32,33,36,38,40,47,50,51,58,59,71,78,79,86,94,98,103,104,107,126,130,131,136,141,142,145,151,155,156,171,175,180,182,184,198]
Berg balance scale	Balance	36	3284 (2225/1059)	81.0	[40,46,63,79,87,93,108,109,110,135,136,138,149,152,167,179,182,183,189,194,198,201,202,203,205,207,208,212,217,221,227,233,235,239,245,247]
Tinetti test	Static and dynamic balance	35	4895 (3541/1332)	81.5	[32,74,77,78,85,86,90,92,97,101,107,145,148,154,168,204,206,211,214,215,218,219,220,222,223,224,225,229,231,234,236,238,240,243,248]
Senior Fitness Test *	Physical fitness, specifically; strength, flexibility, agility and aerobic fitness	18	1643 (1003/412)	80.8	[6,55,70,73,79,81,109,125,136,151,161,162,165,167,168,178,182,191]
Physical Performance Test (9 items)	ADL, balance, and gait	4	301 (267/34)	83.6	[80,195,227,259]
Frailty and injuries: cooperative studies of intervention techniques (FICSIT-4)	Balance	4	263 (195/68)	89.1	[147,150,153,192]
Four Test Balance Scale (FTBS)	Balance	2	52 (47/5)	84.5	[91,146]
Groningen Fitness Test for the Elderly (GFE)	Strength, flexibility, agility, balance, dexterity, reaction time and aerobic fitness.	2	335 (266/69)	83.6	[43,147]
Elderly Mobility Scale	ADL, balance, and gait	2	92 (62/30)	82.5	[60,77]
Nursing Home Physical Performance Test (NHPPT)	ADL, strength, and gait	1	178 (115/63)	78.0	[84]
Balance Evaluation Systems Test	Balance	1	49 (30/19)	77.8	[207]
Mini-Balance Evaluation Systems Test (14-item test)	Balance	1	49 (30/19)	77.8	[207]
Brief-Balance Evaluation Systems Test (14-item test)	Balance	1	49 (30/19)	77.8	[207]
Fullerton Advanced Balance (FAB) Scale	Static and dynamic balance	1	36 (19/17)	No data	[209]

Notes: * In 9 studies, the test battery was partially applied; ADL, activities of daily living.

**Table 3 ijerph-19-05058-t003:** Physical and motor tests applied in the reviewed studies.

Test	N (Females/ Males)	Mean Age	Type of Studies	References
** *Strength tests* **				
Hand grip strength test	13,981 (9720/4112)	82.5	RCT (35) Cross-sectional (45) Pilot study (5) Cohort study (4) Quasi-experimental (5) Prospective study (3)	[29,30,32,35,38,41,43,44,45,47,48,49,50,53,57,64,65,69,70,72,73,74,76,77,78,79,80,83,85,86,87,88,90,98,101,104,106,107,112,113,115,116,117,118,119,120,121,122,123,126,127,128,130,132,133,135,137,138,140,143,144,147,148,150,152,153,155,156,157,158,160,164,166,167,168,169,170,171,172,173,174,175,176,177,179,180,181,182,183,184,186,187,189,190,192,197,200]
Five Times Sit to Stand Test	7083 (4927/2129)	83.2	Cross-sectional (17) RCT (16) Cohort study (10) Pilot study (4)	[10,29,32,33,35,36,38,40,46,47,50,51,58,59,62,65,70,71,73,78,79,86,92,94,96,98,103,104,107,126,130,131,136,141,142,145,150,151,155,156,158,171,175,180,182,184,198]
30 s sit to stand test	3251 (2314/937)	84.5	RCT (14) Cross-sectional (15) Cohort study (2) Pilot study (2)	[31,33,37,52,54,57,63,93,95,97,102,110,116,118,120,122,125,135,143,146,147,149,153,159,163,164,170,173,181,185,187,193,196]
Handheld dynamometer tests	3685 (2774/884)	82.8	Cross-sectional (14) RCT (8) Cohort study (6) Pilot study (3) Quasi-experimental (1)	[32,34,42,43,44,66,67,68,78,82,86,89,93,96,99,100,105,107,111,123,124,129,134,145,147,152,153,154,163,183,188,192]
Arm Curl test	2937 (1915/803)	81.6	RCT (14) Cross-sectional (10) Quasi-experimental (4) Cohort study (4)	[6,55,59,70,73,79,81,91,95,97,109,116,122,125,136,138,143,148,151,160,161,162,165,167,168,170,178,179,181,182,191,196]
Isokinetic Dynamometer (different models)	663 (601/62)	79.9	RCT (6) Cross-sectional (4)	[31,56,61,75,80,90,95,181,186,195,199]
1 RM test (all protocols)	591 (378/213)	86.5	Cross-sectional (4) RCT (3)	[39,83,88,120,153,192,193]
Peak Flow Meter analysis	482 (279/203)	82.1	RCT (3) Cohort study (1)	[160,170,172,179]
Back/leg dynamometer evaluation	158 (89/69)	77.8	RCT (1) Cross-sectional (1)	[108,194]
Leg extension test (GFE)	335 (266/69)	83.6	RCT (1) Cross-sectional (1)	[43,147]
10 RM test	10 (10/0)	86.2	RCT (1)	[114]
Seated Medicine Ball Throw test	31 (17/14)	78.9	Cross-over study (1)	[83]
Lower limb power test	41 (17/24)	69.8	Cohort study (1)	[115]
Cyklotren device	20 (6/14)	76.7	RCT (1)	[97]
Fatigue test–Grip Work	662 (484/178)	83.2	Cohort study (1)	[86]
Test of toe grip strength	35 (23/12)	82.1	Cross-sectional (1)	[139]
Countermovement jump test	31 (17/14)	78.9	Cross-sectional (1)	[83]
11-step stair-climbing test	45 (33/12)	83.8	Pilot study (1)	[57]
8 RM test	15 (9/6)	84.0	Pilot study (1)	[60]
Sitting-rising test	38 (18/20)	73.4	Cross-sectional (1)	[140]
Sit-to-stand (one time) test	178 (115/63)	78.0	RCT (1)	[84]
** *Balance Tests* **				
Tandem test	6239 (4495/1744)	82.8	Cross-sectional (18) RCT (15) Cohort Study (10) Pilot study (2)	[10,29,32,33,36,38,40,43,47,49,50,51,58,59,71,74,78,79,86,94,98,100,103,104,107,108,126,130,131,136,140,141,142,145,151,155,171,175,180,182,184,198,224,233,237]
Semi-Tandem test	6600 (4631/1969)	83.5	Cross-sectional (18) RCT (12) Cohort study (7) Pilot study (2) Quasi-experimental study (1)	[10,29,32,33,36,38,40,47,50,51,58,59,71,78,79,86,94,98,103,104,107,126,130,131,136,140,141,142,145,151,155,156,171,175,180,182,184,198,224,237]
Feet Together test	6536 (4572/1954)	83.5	Cross-sectional (17) RCT (12) Cohort study (7) Pilot study (2) Quasi-experimental study (1)	[10,29,32,33,36,38,40,47,50,51,58,59,71,78,79,86,94,98,103,104,107,126,130,131,136,140,141,142,145,151,155,156,171,175,180,182,184,198,237]
Berg balance scale	2922 (1966/956)	81.1	RCT (17) Cross-sectional (9) Pilot study (6) Exploratory study (1) Cohort study (2) Quasi-experimental (1)	[40,46,63,79,87,93,108,109,110,135,136,138,149,152,167,179,182,183,189,194,198,201,202,203,205,207,208,212,217,221,227,233,235,239,245,247]
Tinetti test	4895 (3541/1332)	81.5	RCT (13) Cross-sectional (10) Longitudinal study (6) Exploratory study (3) Observational study (2) Pilot study (1)	[32,74,77,78,85,86,90,92,97,101,107,145,148,154,168,204,206,211,214,215,218,219,220,222,223,224,225,229,231,234,236,238,240,243,248]
Functional Reach Test -FRT	971 (688/283)	82.7	Cross-sectional (6) Pilot study (4) RCT (6) Cohort study (2) Quantitative study (1)	[11,60,62,64,66,77,96,105,109,157,181,188,209,213,216,225,229,230,237]
One-leg stance	937 (632/305)	81.3	RCT (8) Cross-sectional (4) Quasi experimental (3) Pilot study (3)	[64,70,82,91,96,105,108,111,117,209,210,213,226,230,233,241,243,246]
Postural Sway (force platform or similar)	461 (312/133)	82.9	RCT (3) Cross-sectional (4) Exploratory study (2)	[52,111,138,185,213,230,232,242,244]
FICSIT-4	263 (195/68)	89.1	RCT (3) Cross-sectional (1)	[147,150,153,192]
Turn 360 degrees test	301 (267/34)	83.6	Cross-sectional (3) RCT (1)	[80,195,227,259]
Progressive Romberg test	256 (222/34)	83.2	Cross-sectional (3)	[80,195,259]
4-Stage Balance Test	52 (47/5)	84.5	Cross-sectional (1) Pilot study (1)	[91,146]
Balance board (platform) test (GFE)	335(266/69)	83.6	RCT (1) Cross-sectional (1)	[43,147]
Four Square Step Test	34 (30/4)	83.0	Quasi-experimental (1)	[91]
Balance Evaluation Systems Test	49 (30/19)	77.8	Cross-sectional (1)	[207]
Brief-Balance Evaluation Systems Test	49 (30/19)	77.8	Cross-sectional (1)	[207]
Mini-Balance Evaluation Systems Test	49 (30/19)	77.8	Cross-sectional (1)	[207]
Parallel walk test	117 (76/41)	82.5	RCT (1)	[228]
Fullerton Advanced Balance Scale	36 (19/17)	No data	RCT (1)	[209]
Dynamic Gait Index	22 (no data)	88.2	Cross-sectional (1)	[214]
** *Agility tests* **				
Timed Up and Go test	7212 (4777/2376)	82.2	RCT (35) Cross-sectional (30) Pilot study (8) Cohort study (8) Quantitative study (1)	[6,11,29,31,32,35,45,46,52,57,62,64,65,66,73,76,78,86,87,90,92,93,96,102,105,107,108,115,117,121,136,140,145,146,147,150,151,153,155,158,163,182,183,185,192,194,198,202,203,204,205,208,209,210,211,214,215,216,219,220,221,223,225,226,227,228,229,230,233,235,237,241,243,246,248,249,251,252,253,254,255,256]
8 foot up and go test	1720 (1185/316)	82.5	RCT (8) Cross-sectional (8) Pilot study (1) Quasi-experimental (1)	[49,55,70,74,79,81,97,109,159,161,162,165,167,171,178,191,196,224,257]
** *Gait tests* **				
Gait speed test- SPPB	6469 (4541/1928)	83.7	Cross-sectional (17) RCT (11) Cohort study (6) Pilot study (2) Quasi-experimental study (1)	[10,29,32,33,36,38,40,47,50,51,58,59,71,78,79,86,94,98,103,104,107,126,130,131,136,141,142,145,151,155,156,171,175,180,182,184,198]
4 m walking test	2554 (1728/789)	83.0	RCT (6) Cross-sectional (6) Cohort-studies (3) Case-control (1) Pilot study (2)	[30,38,53,65,73,79,112,129,131,132,133,136,155,158,166,182,184,239]
6 m walking test	1704 (1220/484)	84.0	RCT (7) Cross-sectional (4) Cohort study (4) Pilot study (2) Quasi-experimental (1)	[29,33,37,40,48,54,60,62,63,77,82,84,110,147,149,163,181,241]
10 m walking test	1562 (1171/391)	83.2	RCT (5) Cross-sectional (5) Pilot study (2) Exploratory study (1)	[36,57,68,105,135,187,198,216,217,228,246,249,258]
5 m walking test	1185 (709/476)	86.0	Cross-sectional (4) RCT (2) Exploratory study (1)	[96,106,111,153,177,192,230]
50-foot walk test	301 (267/34)	83.6	Cross-sectional (3) RCT (1)	[80,195,227,259]
Climb One flight of stairs test	301 (267/34)	83.6	Cross-sectional (3) RCT (1)	[80,195,227,259]
Climb Four Flights of stairs test	301 (267/34)	83.6	Cross-sectional (3) RCT (1)	[80,195,227,259]
4.6 m Walking Test	700 (621/79)	81.7	Cross-sectional (2) RCT (1)	[49,74,144]
GAITRite system	81 (56/25)	82.7	RCT (1) Cohort study (1)	[156,260]
Locometrix® gait analysis system	124 (94/30)	83.2	RCT (2)	[219,248]
7 m walking test	24 (18/6)	93.1	Cross-sectional (1)	[39]
3 m Walking Test	70 (42/28)	83.2	Cross-sectional (1)	[35]
7.5 m Walking Test	60 (47/13)	85.5	Pilot study (1)	[259]
8 m Walking test	226 (184/42)	81.6	Cross-sectional (1)	[43]
Figure of 8 walk test	87 (67/20)	87.0	RCT (1)	[150]
10 m Maximal Walking Speed test	31 (25/6)	89.0	Cross-sectional (1)	[250]
Stepping test (repetitive side-stepping tester-TKK5301 TAKEI Co)	40 (36/4)	83.8	Cross-over study (1)	[105]
Step test (BOOMER protocol)	46 (28/18)	82.4	Pilot study (1)	[64]
** *Aerobic capacity tests* **				
6 Min walking test	2486 (1638/629)	82.9	RCT (16) Cross-sectional (10) Pilot study (4)	[6,31,46,55,79,80,81,87,93,95,116,117,119,136,147,150,151,161,163,164,173,181,182,186,189,191,202,259,261,262]
2-Min Step Test	553 (429/124)	81.5	RCT (1) Cross-sectional (5) Quasi-experimental (1)	
Walking endurance test (GFE)	335 (266/69)	83.6	RCT (1) Cross-sectional (1)	[43,147]
10-min walk (weellchair) distance test	418 (246/172)	78.3	Cohort study (1) RCT (1)	[120,193]
Cooper test	42 (29/13)	83.21	RCT (1)	[243]
** *Flexibility tests* **				
Back-Scratch test	2111 (1307/585)	80.2	RCT (8) Cross-sectional (8) Quasi-experimental (2)	[43,55,73,79,81,90,100,109,138,160,161,162,165,168,170,178,179,191]
Chair–Sit and Reach test	1703 (970/514)	80.1	RCT (8) Cross-sectional (6) Quasi-experimental (1) Cohort (1)	[55,73,79,81,90,109,138,160,161,162,165,170,178,179,191,246]
Goniometric measures	1231 (861/370)	81.7	RCT (6) Cross-sectional (3)	[65,101,138,160,170,179,188,263,264]
Sit and Reach test	780 (595/185)	80.9	RCT (4) Cross-sectional (2) Exploratory study (1) Quasi-experimental (1)	[43,82,100,108,111,147,159,194]
Lateral Reach test	74 (50/24)	81.0	Cross-sectional (2) RCT (1)	[188,213,226]
Circumduction test (GFE)	335 (266/69)	83.6	RCT (1) Cross-sectional (1)	[43,147]
Thomas test	17 (17/0)	67.0	RCT (1)	[61]
** *ADL tests* **				
Book Lift test	301 (267/34)	83.6	Cross-sectional (3) RCT (1)	[80,195,227,259]
Put on and remove a coat test	301 (267/34)	83.6	Cross-sectional (3) RCT (1)	[80,195,227,259]
Pick up a penny test	301 (267/34)	83.6	Cross-sectional (3) RCT (1)	[80,195,227,259]
Lying to Sitting test	92 (62/30)	82.5	Exploratory study (1) Pilot study (1)	[60,77]
Sitting to Lying test	92 (62/30)	82.5	Exploratory study (1) Pilot study (1)	[60,77]
Sitting to Standing test	92 (62/30)	82.5	Exploratory study (1) Pilot study (1)	[60,77]
Scooping applesauce test	178 (115/63)	78.0	RCT (1)	[84]
Face washing test	178 (115/63)	78.0	RCT (1)	[84]
Dial a telephone test	178 (115/63)	78.0	RCT (1)	[84]
Put On/Take Off Sweater test	178 (115/63)	78.0	RCT (1)	[84]
Write a sentence test	45 (45/0)	84.8	RCT (1)	[227]
Simulated eating test	45 (45/0)	84.8	RCT (1)	[227]
** *Reaction time tests* **				
Reaction time test (GFE)	335 (266/69)	83.6	RCT (1) Cross-sectional (1)	[43,147]
Deary-Liewald Reaction Time Task	91 (63/28)	83.7	Pilot study (2)	[6,109]
Reaction time test (pushing a button as fast as possible)	159 (127/32)	81.6	RCT (1)	[100]
** *Manual dexterity tests* **				
Box and Block Test	489 (323/166)	78.0	RCT (2) Cross-sectional (1)	[100,138,179]
Block transfer test- GFE	335 (266/69)	83.6	RCT (1) Cross-sectional (1)	[43,147]
Purdue pegboard test	52 (33/19)	81.0	RCT (1)	[157]
** *Proprioception tests* **				
Arm Ruler Positioning test	53 (41/12)	85.9	Quantitative study (1)	[11]
Lower limb matching tasks	116 (64/52)	76.6	Cross-sectional (1)	[194]
Knee joint position sense test	42 (25/17)	79 *	RCT (1)	[108]
Weigth Detection Test	53 (41/12)	85.9	Quantitative study (1)	[11]
** *Paratonia tests* **				
Paratonia Assessment Instrument, PAI	79 (62/17)	84.2	Cross-sectional (1)	[265]

Notes: RM, repetition maximum; RCT, randomized controlled trial; ADL, activities of daily living; SPPB, Short Physical Performance Battery; GFE, Groningen Fitness Test for the Elderly; *, median.

**Table 4 ijerph-19-05058-t004:** Methodological quality of the studies that assessed the tests’ psychometric properties.

	Content Validity	Structural Validity	Internal Consistency	Cross-Cultural Validity	Reliability	Measurement Error	Criterion Validity	Construct Validity	Responsiveness
Hand-held dynamometer [68]	NA	NA	NA	NA	Very good	Inadequate	NA	Adequate	NA
Hand-held dynamometer [34]	NA	NA	NA	NA	Very good	Adequate	NA	NA	NA
Tinetti Test [222]	NA	NA	NA	NA	Adequate	NA	Very good	NA	Inadequate
10 m maximal walking speed [250]	NA	NA	NA	NA	Adequate	Adequate	NA	NA	NA
Berg balance Scale [63]	NA	NA	Very good	NA	Very good	Very good	NA	Very good	NA
Berg balance Scale [207]	NA	NA	NA	NA	Very good	Very good	Very good	NA	Adequate
30 s chair stand test [63]	NA	NA	NA	NA	Adequate	Adequate	NA	NA	NA
6 m walking test [63]	NA	NA	NA	NA	Adequate	Adequate	NA	NA	NA

Notes: NA, not applicable.

**Table 5 ijerph-19-05058-t005:** Reliability, validity, and summary of the tests’ psychometric properties.

Assessment Method	Reliability	Validity	Findings
Hand-held dynamometer	Test-retest reliability (ICC): 0.97 [68]; 0.60–0.87 [34] Absolute reliability (SEM/MDC95): 6.17–37.99/8.80–29.90 [34]	-	Reliable method in older adults with dementia [68]. Reliable to assess isometric strength of several muscle groups [34].
Tinetti test	Inter-rater reliability (ICC): 0.97 [222]	Predictive validity: sensitivity, 70–85%; specificity, 51–61% [222]	In populations with moderate to severe dementia, this method is hampered by feasibility problems. Its application in clinical practice cannot therefore be recommended, despite an acceptable predictive validity [222].
10 m maximal walking speed	Test-retest reliability (ICC): 0.86 [250]	-	The test has high reliability in institution-dwelling older people aged 65 years and older, with several different diagnoses [250].
Berg balance Scale	Inter-rater reliability (ICC): 0.99 [63]; 0,99 [207] Test-retest reliability (ICC): 0.89 [207] Absolute reliability (SEM/MDC_95_): 0.97/1.92 [63]; 3.8/10.5 [207];	Construct validity (Cronbach’s α): 0.95 [63] Criterion validity (sensitivity/specificity): 0.94/0.55 [207];	Excellent relative inter-rater reliability of the BBS, as well as high internal consistency, in a population of nursing home residents with mild-to-moderate dementia [63]. The test presented similar reliability, reproducibility, and validity [207].
30 s chair stand test	Inter-rater reliability (ICC): 1.0 [63] Absolute reliability (SEM/MDC_95_): 0/0 [63]	-	Excellent relative inter-rater reliability [63].
6 m walking test	Inter-rater reliability (ICC): 0.97 [63] Absolute reliability (SEM/MDC_95_): 0.03/0.06 [63]	-	Excellent relative inter-rater reliability [63].
Balance Evaluation Systems Test	Inter-rater reliability (ICC):0.99 [207] Test-retest reliability (ICC): 0.95 [207] Absolute reliability (SEM/MDC_95_): 5.6/15.6 [207];	Criterion validity (sensitivity/specificity): 0.83/0.61 [207];	Good reliability, reproducibility, and validity [207].
Mini-BESTest	Inter-rater reliability (ICC): 0.99 [207] Test-retest reliability (ICC): 0.93 [207] Absolute reliability (SEM/MDC_95_): 1.8/4.9 [207];	Criterion validity (sensitivity/specificity): 0.78/0.71 [207];	Good reliability, reproducibility, and validity [207].
Brief-BESTest	Inter-rater reliability (ICC): 0.99 [207] Test-retest reliability (ICC): 0.94 [207] Absolute reliability (SEM/MDC_95_): 1.4/4.0 [207]	Criterion validity (sensitivity/specificity): 0.94/0.58 [207];	Good reliability, reproducibility, and validity [207].
Weight Detection Test	Test-retest reliability (ICC): 0.84 [11] Absolute reliability (SEM/MDC_95_): 1.0/2.8 [11]	-	Excellent test-retest reliability and acceptable measurement precision [11].
Arm Ruler Positioning Test	Test-retest reliability (ICC): 0.87 [11] Absolute reliability (SEM/MDC_95_): 0.3/0.9 [11]	-	Excellent test-retest reliability and acceptable measurement precision [11].
Functional Reach Test	Test-retest reliability (ICC): 0.85 [11] Absolute reliability (SEM/MDC_95_): 1.5/4.0 [11]	-	Excellent test-retest reliability and acceptable measurement precision [11].
Timed Up and Go test	Test-retest reliability (ICC): 0.99 [11] Absolute reliability (SEM/MDC_95_): 0.5/1.5 [11]	-	Excellent test-retest reliability and acceptable measurement precision [11].

## Data Availability

This is a review paper, data are presented throughout the text.

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
