# Peer review of "Physical and Motor Fitness Tests for Older Adults Living in Nursing Homes: A Systematic Review"

_ijerph, 2022, doi:10.3390/ijerph19095058_

Round 1
Reviewer 1 Report
I would like to thank the editorial board for the opportunity to review the current manuscript that was titled “Physical and motor fitness tests for older adults living in nursing 2 homes: a systematic review”.
The most common physical component assessed was muscle strength, balance, and agility. This review also described the most used assessment tests for each physical/motor component, whereas manual dexterity and proprioception have been considered less.
Comments
General
Overall, the manuscript is well written with clearly presented results. I have two minor comments that I would like to see addressed.
Introduction
The introduction provides a background of the aim of the literature review. The concerns I had while reviewing the manuscript have been raised in the limitations section.
Methods
L76 – update the word “institutionalized”
Results
Tables are clearly presented and easy to identify how many studies were included in each category. Nice to see the inclusion of male/female number of subjects in each category
Reviewer 2 Report
Title: Physical and motor fitness tests for older adults living in nursing homes: a systematic review
Overall:
Generally, the paper is well written and informative. The writers do a great job of explaining the purpose and relevance of research in this area and the importance of this systematic review. Even though there were a few minor grammatical errors dispersed throughout the paper, overall the paper was thorough and a nice read.
Abstract:
The abstract is well written and clearly states the purpose, results, and expectations for this systematic review.
Introduction:
Line 33- Consider citing fact about average age expectancy
Line 49- Consider going into more depth about what “capacities” you are referring to
Materials and Methods:
Line 66 – were the new 2020 guidelines used? If so please note
Line 73 (and in abstract) searched from Jan 2002 – etc. do you mean articles published during that time were searched? As written it sounds like the authors were searching during that time
Line 108-112 – Cite what data extraction tool was used
Line 114 – “in this revision”? your point isn’t clear what that means, is this a revision of a previous article?
Results:
Few grammatical errors but still well written and provided the necessary information to understand the results of the systematic review.
Line 217 – identify what the balances tests do – and cite each. Nothing is cited in this paragraph.
Line 222 – describe TUG, also lacking proper referencing.
Line 229 – describe this tests – also a lack of references in this paragraph
Line 231, 226, 240 also - All paragraphs in this section lack citations and descriptions of major tests
Section at 242 – (and where previously noted) it’s confusing to the reader that you included the quality of the tests. Please provide a better explanation of why some were included and others not – right now it seems as if it was just because it was easy.
EG line 295 – explain here as well
Discussion:
See comments in the results section about describing the tests. That would fit in the discussion as well.
Make sure the discussion section is not simply re-telling your results. More discussion is needed on the importance of information and application. Don’t just repeat what you found.
There also are very few citations in the discussion overall. Anything that is not common knowledge must be cited. For example, like 349, you found 7 studies that used methods of testing flexibility – but there were no citations included. Correct throughout the paper and esp the discussion.
Line 281-283- “Although, only five…this review” This sentence does not read well consider changing the wording.
Line 283-286- “The most popular…(relative strength to body mass)” This sentence does not read well consider changing the wording.
Line 290- Ability to “transfer” what? Be specific
Limitations:
Line 406-410- Consider adding “for older adults in nursing homes” to be more clear about what population this systematic review is providing relevant evidence on
References:
Well done. Very thorough.
Figures and Tables:
Table 3: Consider other ways in listing which references apply to which test.
Tables – suggest using a more standard format of including the author name and year rather than just the reference #
Reviewer 3 Report
This systematic review deals with Physical and motor fitness tests for older adults living in nursing homes. While the systematic review is properly performed, the numbers of papers included prevent to provide a clear take home message.
The authors included RCT, Cross-sectional, Cohort study, Pilot study and Quasi-experimental study. Since the RCT have the higher level of research strength, the authors should have focused on including only the RCT.
Then, they should have evaluated the quality of the RCT using a dedicated critically appraised sheet and then they should have evaluated the risk of bias associated to draw clear conclusions.
This review is a full list of data without clear take home message. Please focus on RCT, reduce the length and provide a take home message that is validated
Reviewer 4 Report
Physical and motor fitness tests for older adults living in nursing homes: a systematic review
This is a very good review study. Authors established an interesting research question and manage to gather a large part of the scientific literature in order to present the most frequently used physical and motor fitness tests in older adults. This is a very strong review and I comment the authors for the effort.
Overall, the review is already in a good state. Authors have done a great writing work in all paragraphs, tables are crystal clear and the main question of the review is certainly answered. On the other hand, reading this manuscript generates more questions to the readers such as how the systematic exercise may affect these measurements, or how a trainer or a health-care professional can choose which tests to apply in this population. Thus, I suggest to authors maybe look into to these questions for future research.
Thus, I have only minor comments for this review. This is an apt and well-written text, with an enormous reference list and very strong findings.
Introduction: Please, add a few lines about the overall effect of systematic exercise on the health status of elderly, and the potentially effect on these physical and motor fitness tests.
Methods: Was there a separation between male and female participants depending on the physical and fitness tests? Was there a test more frequently used in males and females?
There is a question here: How authors exclude or include the studies with participants with chronic non-communicable diseases?
Paragraph 3.2: I suggest using bullets here for each small paragraph. It would be much easier for the reader.
Tables are very good.
Discussion:
Ln 283-297: Generally, muscle mass has an important role in all fitness tests in elderly. Strength is a vital component for elderly because of the strong link with muscle mass. Please, add a few lines here about the connection between strength and muscle mass.
Future research should be pointed out after limitations.
Conclusions:
Please add the main findings here again for summarizing the review.
Well done.
